# OpenReview forum: "Sampling from Your Language Model One Byte at a Time"
_ICML.cc/2025/Workshop/TokShop — TokShop_

### Official Review · Reviewer_KaTQ · 2025-06-08
**An Elegant and Efficient Solution to the Prompt Boundary Problem with Novel Applications**

**Rating:** 6
**Confidence:** 4

**Review:**

This paper addresses the "Prompt Boundary Problem" (PBP), a well-known issue where language models produce suboptimal completions when a prompt ends on a prefix of a valid token. The authors propose ByteSampler, an efficient and exact algorithm that enables byte-level sampling from any BPE tokenizer-based language model at inference time. The core of the method is the "Valid Covering Tree," a data structure that represents all valid tokenizations of a given byte prefix. Crucially, the paper demonstrates two novel and powerful applications enabled by this method: (1) creating ensembles of LMs with different tokenizers and (2) performing "proxy-tuning" to transfer post-training capabilities between models at inference time.

Strengths:
Methods and Evaluation: The experimental design is rigorous and comprehensive. The authors convincingly demonstrate the efficiency of their method against other exact approaches (Table 2), its correctness in preserving the model's distribution (Tables 3 and 5), and its superiority in next-character prediction over strong heuristic baselines (Tables 4 and 6). The proof-of-concept experiments for ensembling and proxy-tuning successfully validate the practical utility of ByteSampler on downstream tasks.

Weaknesses
1. Pretokenization: The handling of complex pretokenization rules is relegated to the appendix and relies on bespoke handlers, which feels less general than the core BPE algorithm.
2. Limited Scope: The discussion is confined to BPE tokenizers, with limited exploration of how the approach might extend to other schemes like Unigram or WordPiece.

---

### Official Review · Reviewer_FyRJ · 2025-06-08
**Good paper but requires some attention**

**Rating:** 7
**Confidence:** 5

**Review:**

Overall the strengths outweigh the weaknesses, but please do address the weaknesses.

$\textbf{Summary} $

This paper proposes $\textit{ByteSampler}$, an algorithm to eliminate the Prompt Boundary Problem in Language Models (LMs). Their solution falls within the "exact" solution methodologies proposed in previous works. They show that their method stands out when compared to contemporaries, mainly due to its lower computational cost. The method is shown to incur minimal loss in language modeling tasks, making it a viable solution. Their approach involves converting the LM into a byte-level LM and constructing a Valid Covering Tree to predict the next best byte. In addition to experimentally demonstrating the success of their novel approach, they also elaborate on the applications of their methodology. Specifically, since the models are converted to byte-level LMs, this enables byte-level ensembling of multiple models and byte-level proxy tuning.

$\textbf{Strengths}$
1. The approach is novel, and the authors have shown that they have considered essential prior work.
2. Experimentally shows that, out of the available "exact" methods, their proposed methodology has a clear advantage in computational cost while having minimal language modeling degradation. (there is an associated weakness with this, please refer to point \#2 of the Weakness section)
3. Rather than just claiming the applications, they also experimentally show that their method could be used for Byte-Level ensembling and Byte-Level proxy tuning, both showing promising results using their approach.
4. Proper information is provided throughout the paper in the form of Background and Appendixes. (please also look at the Suggestion to improve the paper structure).

$\textbf{Weakness}$
1. At times, the authors have failed to provide evidence for some of their claims. Unless a claim is proven within the work, supporting evidence must be provided through prior research. For example, the claim "Longer tokens, however, can distort the sampling distribution if the boundary between the prompt and its completion is not carefully handled." (Lines 29--32).
2. In Tables 3 \& 5, why haven't the authors included other "exact" methods  for comparison against the proposed method and the baseline Plain BPE?
3. Authors also need to show results of the chat-aligned (instruct models) counterparts of the models compared (OLMo2 for English and QWEN3 for Chinese) using prompts to request the model to complete the PBP text. So far, the results that have been presented rely solely on properties of a base model that performs next-token prediction.
4. Why consider and stop with only 2 models? It is essential to show that this method extends well to other models as well (e.g., Mistral, Phi, LLAMA).

$\textbf{Suggestion to improve paper structure}$
1. The abstract is too long. There is verbosity in the abstract; I believe it could be made more concise.
2. I feel that some of the text in the Introduction section actually belongs to the Background section. (For example: lines 29--54 in the right-hand side column)
3. It also gives the impression that this could be a journal paper, as the authors are covering a large scope within 9 pages. It could easily be extended into a journal paper.
4. Include Limitations section.

---

### Decision · Program_Chairs · 2025-06-10

Accept